# Improving Photocatalytic Stille Coupling Reaction by CuPd Alloy-Doped Ordered Mesoporous TiO2

Ting Tang [1], Lehong Jin [1], Wei Chai [2], Jing Shen [3], Zhenmin Xu [3,*] and Haifang Mao [3,*]

1    School of Public Health, Hangzhou Normal University, Hangzhou 311121, China
2    Department of Chemical Engineering, Zaozhuang Vocational College, Zaozhuang 277800, China
3    School of Chemical and Environmental Engineering, Shanghai Institute of Technology,
     Shanghai 201418, China
*    Correspondence: zhenminxufy@163.com (Z.X.); mhf@sit.edu.cn (H.M.)

**Abstract:** Rational surface engineering of noble metal-doped photocatalysts is essential for the efficient conversion of solar energy into chemical energy, but it is still challenging to perform. Herein, we reported an effective strategy for structuring alloyed CuPd (CP) nanoclusters on the ordered mesoporous $TiO_2$ (CPT) by a pore confinement effect. The resultant CPT exhibited an extraordinary photocatalytic activity during Stille reaction under visible light. The X-ray photoelectron spectroscopy spectra, the field emission scanning electron microscope (FESEM) images, and the aberration-corrected high-angle annular dark scanning transmission electron microscopy (HAADF-STEM) images demonstrated that CP nanoclusters were anchored in the mesoporous pore wall of $TiO_2$, and the atomic ratio as well as densities of CP could be precisely modulated via the coordination configuration. As the atomic ratio of CP to $TiO_2$ increased to a certain extent, their photocatalytic activity during Stille reaction increased. A mechanistic investigation suggested that the CP alloy could absorb visible light and its conduction electrons gained energy, which were available at the surface Pd sites. This allowed the Pd sites to become electron-rich and to accelerate the rate-determining step of the Stille reaction. As a result, the efficiency of the photocatalytic Stille coupling reaction was extraordinary enhanced.

**Keywords:** photocatalytic; Stille coupling reaction; CuPd alloy; mesoporous $TiO_2$





## 1. Introduction

The Stille cross-coupling reaction, as one of the most powerful C-C bond-forming reactions, has been widely used in natural products, pharmaceutical and polymer syntheses due to the stability of organostannanes, the broad scope of reaction substrate, the functional group tolerance as well as the high chemoselectivity [1–3]. The current Stille cross-coupling reaction is traditionally carried out with the aid of various palladium-based homogeneous catalytic systems. However, this approach has suffered from high costs and difficulty in the separation and recovery of Pd catalysts [4,5]. In recent years, Pd nanoparticles that are based on heterogeneous systems have been developed as candidate catalysts that could operate under mild conditions and be separated and recovered more easily, although their catalytic activity is generally inferior to that of the homogeneous Pd catalysts [6]. In addition, the Stille cross-coupling reaction involves redox processes, including oxidative addition, metal migration, isomerization, and reduction elimination. The key to redox processes is the transfer of electrons between metal catalysts and organic substances, which requires energy input to overcome the high reaction energy barrier to activate the C-X bond and to enable the effective conversion of organic electrophiles and organostannanes to desirable products [7–10]. Nevertheless, a high temperature not only increases the possibility of the formation of undesirable products but also can be detrimental to the stability and reusability of catalysts. Thus, efficient catalytic systems that are based on

recyclable catalysts and green energy sources and that can carry out these reactions at an ambient temperature are highly desirable but remain a significant challenge.

As an inexhaustible green energy source, sunlight has been increasingly used in chemical reactions [11,12], and numerous studies have demonstrated that photoexcited electrons can also trigger C−C coupling at ambient conditions in the presence of photocatalysts [13–15]. For example, Xiao et al. discovered that Au/Pd alloy can strongly absorb and gain energy from visible light; the availability of photoexcited electrons with high energy densities on the Pd surfaces not only increases their affinity for the reactant molecules but also enhances their intrinsic ability to activate the reactant molecules. Thus, the conversion of the Suzuki cross-coupling reaction is efficiently enhanced at a mild temperature [16,17]. Although Xiao et al. have achieved satisfactory results, the application of gold and palladium alloy is limited by its complex preparation process and high price. Semiconductor-supported palladium and its alloys are considered promising photocatalysts for C-C formation due to their effective absorption of visible light, good durability, capacity for multivalent binding to substances, and cost-effectiveness. However, it is still a challenge to precisely control the particle size and distribution of Pd and its alloys.

Mesoporous $TiO_2$ usually exhibits high photocatalytic activity, owing to its improved reactant diffusion and light absorbance [18–21]. Additionally, the open pore structure of the catalyst is crucial because it promotes the adsorption and diffusion of target molecules; the enlarge BET surface areas can provide more active sites, and more importantly, the multiple scattering of light that occurs in the mesopore will also improve light harvest [22]. Herein, we reported the synthesis of a CuPd alloy that was embedded into $TiO_2$ via a pore-confined strategy by in situ surfactant-assisted self-assembly (Figure 1). In this process, the uniform P123/ titanium oligomer composite micelles were formed and worked as the building blocks to enable the self-assembly of micelles into the ordered mesostructure, where the oligomer not only provided a reinforced structure to retain the highly ordered mesostructure but also acted as a cross-linked network to maintain the confinement of other metal ions. After crystallization, the isolated CuPd atoms were confined to the pore wall of the mesostructured $TiO_2$ [23]. As a result, the size and distribution of the CuPd alloy (CP) could be easily controlled. Meanwhile, the mesoporous structure could be tailed and engineered. The CP modified meso$TiO_2$ (CPT) exhibited a high photocatalytic activity during Stille cross-coupling reaction, owing to the synergetic effects from CuPd and the ordered mesoporous $TiO_2$.

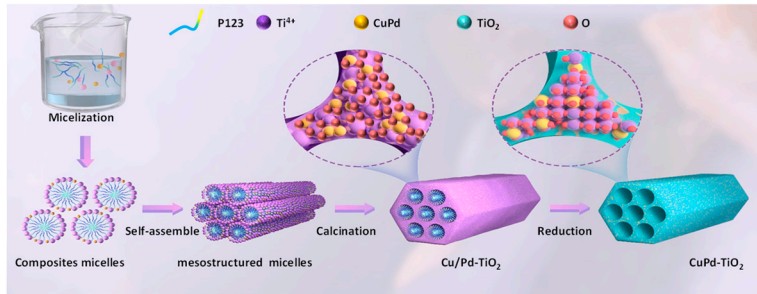

**Figure 1.** Illustration of the synthesis of a CuPd-$TiO_2$(CPT) catalyst.

## 2. Experimental

### 2.1. Chemicals and Materials

Tributyl(vinyl)tin ($C_{14}H_{30}Sn$), Tributyl(Phenyl)Stannane ($C_{18}H_{32}Sn$), Iodobenzene ($C_6H_5I$), 4-Iodotoluene ($C_7H_7I$), 4-Iodoanisole ($C_7H_7IO$), 4-Iodonitrobenzene ($C_6H_4INO_2$), Cupric acetate ($C_4H_6CuO_4$), Palladium(II) acetate (Pd(OAc)$_2$), Titanium(IV) butoxide ($C_{16}H_{36}O_4Ti$), Titanium(IV) Chloride (TiCl$_4$), Potassium Carbonate ($K_2CO_3$), Sodium Hydroxide (NaOH), Triethylamine (TEA, $C_6H_{15}N$), N-Ethyldiisopropylamine (DIPEA, $C_8H_{19}N$), Ethanol (EtOH, $C_2H_5OH$), Acetonitrile (MeCN), 1,4-Dioxane ($C_4H_8O_2$), N,N-Dimethylformamide (DMF, $C_3H_7NO$), and Dimethyl sulfoxide (DMSO, $C_2H_6OS$) were pur-

chased from Sinopharm Chemical Reagent (Shanghai, China). Pluronic P123 ($EO_{20}PO_{70}EO_{20}$, Mw = 5800 g/mol) was obtained from Sigma-Aldrich. All chemicals were used without further purification.

### 2.2. Preparation of Ordered Mesoporous TiO₂

$TiO_2$ with ordered mesoporous was synthesized as follows. Together, 1.0 g P123, 1 mmol $TiCl_4$, 1 mmol $C_{16}H_{36}O_4Ti$, and 20 mL EtOH were mixed and vigorous stirred for 1 h. Then, the resultant solution was transferred into petri dishes (15 × 15 mm) and dried in an oven at 40 °C for 24 h and then at 80 °C for another 24 h. Finally, the ordered mesoporous $TiO_2$ was obtained via calcination at 350 °C in $N_2$ for 2 h and 400 °C in air for 3 h.

### 2.3. Preparation of Ordered Mesoporous TiO₂ with Cu or/and Pd Doped

Pd/$TiO_2$ (PT) with ordered mesopores was fabricated using the following procedure. Specifically, 1.0 g P123, 1 mmol $TiCl_4$, 1 mmol $C_{16}H_{36}O_4Ti$, 0.05 mmol $Pd(OAc)_2$, and 20 mL EtOH were mixed and vigorous stirred for 1 h. The resultant solution was transferred into petri dishes (15 × 15 mm) and dried in an oven at 40 °C for 24 h and then at 80 °C for another 24 h. Finally, gray powder was obtained via calcination at 350 °C in $N_2$ for 2 h, 400 °C in air for 3 h, and 280 °C in $H_2$ for 3 h, which was denoted as Pd/$TiO_2$ (PT).

Similar procedures were employed to fabricate Cu-$TiO_2$ and CuPd-$TiO_2$, except that $Pd(OAc)_2$ was replaced in whole or in part by $Cu(OAc)_2$. The obtained samples were denoted as CT and CPT-X, respectively, where X was the molar ratio of Cu to Pd. In this paper, CPT-0.2, CPT-0.6, CPT-1.0, CPT-1.5, and CPT-2.0 were synthesized and their photocatalytic properties were studied, and the molar ratio of Cu to Pd for each sample was determined using ICP and XPS (Table S2).

### 2.4. Photocatalytic Stille Coupling Reaction

CPT (20 mg) was dispersed in 2 mL EtOH, followed by the addition of 0.1 mmol of aryl halides, organostannanes ($C_{14}H_{30}Sn$ or $C_{18}H_{32}Sn$), and 0.5 mmol of $K_2CO_3$, in a Pyrex reactor equipped with a rubber septum. The reaction solution was irradiated with a 300 W Xenon lamp (λ > 420 nm) at room temperature. After irradiation for a certain period time, the reaction mixture was filtered and analyzed by a high-performance liquid chromatograph (Agilent 6410 Series Triple Quad), which was equipped with an Agilent C18 reverse-phase chromatographic column and a UV detector (wavelength of 280 nm). The mobile phase was 55:45 acetonitrile/water (flow rate of 1 mL/min and injection volume of 5 μL). The conversion of aryl halides and the yield of corresponding products were determined using trimethylbenzene as the internal standard sample.

## 3. Results and Discussion

### 3.1. Structural Properties of CPT

The X-ray diffraction (XRD) patterns in Figure 2a showed seven diffuse peaks which could be well-matched to anatase. The negligible variation in the position and intensity of those diffuse peaks and undiscovered peaks, for both CuPd alloy and their monomers, indicated well-retained phases and highly dispersed crystalline nanoclusters after the introduction of CuPd into the skeleton of the mesoporous $TiO_2$ [24,25]. The SAXS patterns of the CPTs in Figure 2b showed two well-defined scattering peaks assigned to the 10 and 11 reflections of the hexagonal symmetry (*p6m*), implying a highly ordered mesostructure [26]. Type IV isotherms with a capillary condensation at a middle pressure were observed in Figure 2c, suggesting all CPTs possessed large mesoporous size. The H1-type hysteresis loops were found for all CPTs, suggesting uniform cylindrical mesochannels [27]. In addition, pore volumes and BET surface areas of CPT-0.2, CPT-1.5, and CPT-2.0 were estimated to be 0.46, 0.32, and 0.30 $cm^3$/g, and 180, 164, and 155 $m^2$/g, respectively, with a pore size distribution of 4.1–3.2 nm (Table S1). The high BET specific surface areas, large pore volumes, and uniform pore size distribution (Figure 2d, Table S1) determined the

variation in the atomic ratio of CuPd and slightly affected the mesostructure. These results were indicative of the fact that the atomic CuPd alloy was anchored into the skeleton of the mesostructure of TiO$_2$ via a pore confinement effect. This enabled the CPTs to have open and ordered mesopores.

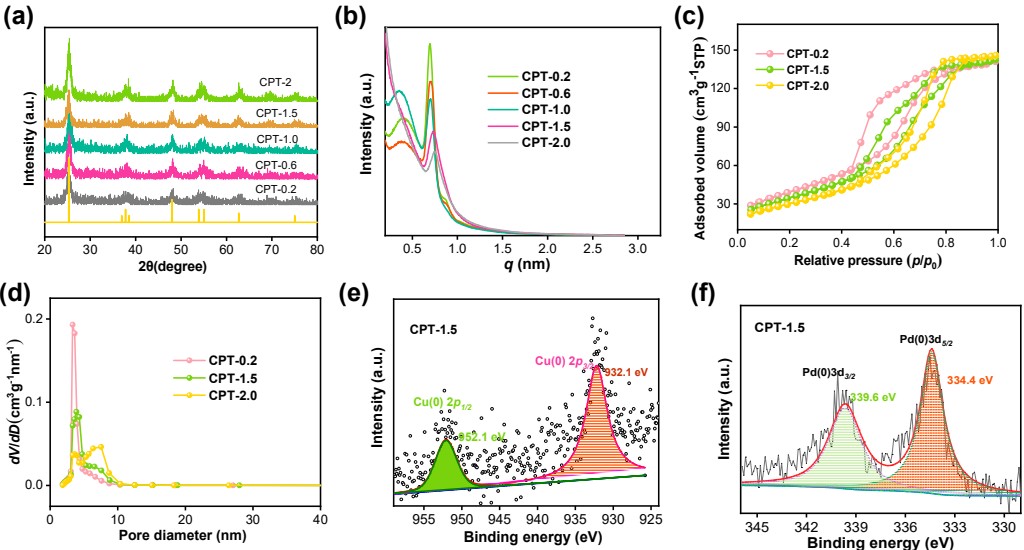

**Figure 2.** Structural analysis of CPT-X. (**a**) wide-angle (**b**) SAXS patterns of CPT with different Cu/Pt ratio, (**c**) N$_2$ sorption isotherms, (**d**) pore-size distribution curves of CPT-0.2, CPT-1.5, and CPT-2.0, (**e**) Cu 2p, and (**f**) Pd 3d XPS spectra of CPT-1.5.

The electronic configuration of CPT was explored using XPS. The high-resolution XPS spectra of Ti revealed that the intrinsic values of Ti 2p$_{3/2}$ and Ti 2p$_{1/2}$ had a slightly negative shift (Figure S1), signifying that the introduction of CuPd was accompanied by the formation of oxygen vacancy (*Ov* bulk). This hypothesis was further verified by the high-resolution XPS spectra of O (Figure S2). In the case of Cu and Pd XPS spectra (Figure 2e,f), the characteristic peaks located at 952.1 eV and 932.1 eV in the Cu XPS spectra of CPT could be attributed to Cu 2p$_{1/2}$ and 2p$_{3/2}$ of Cu(0), and the peaks at 339.6 eV and 334.4 eV in the high-resolution XPS of Pd could be assigned to Pd 3d$_{3/2}$ and Pd 3d$_{5/2}$ of Pd in the CPT, respectively [28]. In addition, the atomic ratio as determined by the XPS was basically consistent with the results of the ICP test (Table S2), implying the atomically dispersed CuPd alloy was embedded into the wall of the mesostructure via a pore confinement effect.

The field emission scanning electron microscope (FESEM) images of the CPTs showed a typical bulk configuration with a size of approximately 2 mm (Figure S3). The high-resolution FESEM images exhibited ordered and open mesoporous arrays in large domain, and no apparent CuPd nanoparticles on the external surfaces or inside of the pore channels were found (Figure S4). The mesostructures of the sampled CPTs were further confirmed using transmission electron microscope (TEM). As shown in Figure S5, all of them possessed a 2D hexagonal mesoporous structure with an average diameter of 3.0~4.0 nm and an average thickness of 3.0~4.0 nm for the pore walls. These results were in agreement with the results of the SASX. Apparently, isolated CuPd alloy (bright spots) with a size of 2.5 nm could be found on the aberration-corrected high-angle annular dark scanning transmission electron microscopy (HAADF-STEM) image (Figure 3). These bright spots were confined to the pore wall of TiO$_2$. Moreover, the corresponding energy-dispersive X-ray spectroscopy (EDX) elemental mapping images demonstrated that CuPd was uniformly distributed on TiO$_2$, which had an ordered mesoporous structure.

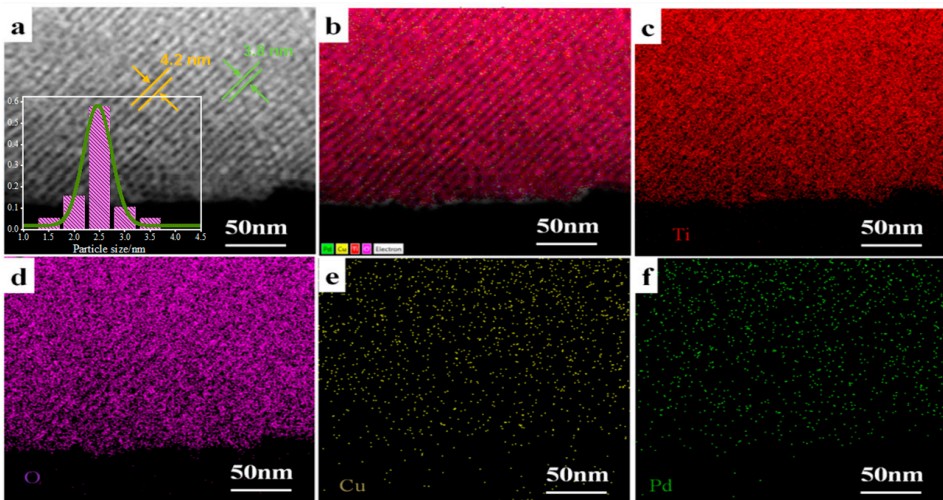

**Figure 3.** (**a**) HAADF-STEM image for CPT-1.5 and (**b–f**) the corresponding EDS elemental mapping images.

### 3.2. Photocatalytic Stille Coupling Reaction

The photocatalytic activity of the CPT was conducted using Stille coupling reaction under visible light at room temperature, as shown in Figure 4a. The CPT was added into an ethanol (EtOH) solution of Iodobenzene and Tributyl(vinyl)tin with $K_2CO_3$ as a base. This mixture was stirred for 1 h under visible light irradiation using Xe lamp (equipped with a cut-off filter at 420 nm) to enable Iodobenzene to be converted into the corresponding product styrene. Although Pd is well-accepted as the ideal co-catalyst to activate C-X and achieve C-C coupling, such as the Suzuki and Ullmann reactions [29,30], Pd-doped $TiO_2$ (PT) showed the expected activity toward Iodobenzene conversion (Figure 4b). Significantly, the activity of PT was further improved by introducing Cu; all of the CPTs showed a higher efficiency of Iodobenzene conversion compared with pristine PT under the same condition. In particular, the photocatalytic activity of the CPT increased at first and then slightly decreased with an increasing amount of Cu, which presented a volcano-type trend with an increase of Cu to Pd ratio (Figure S6). This result suggested the CuPd ratio was an important factor that was affected the photocatalytic performance of CPT, and the optimal CPT could almost realize the complete conversion of aryl halides to their corresponding products within 3 h (Figure 4c). This result was superior to the results previously reported for heterogeneous Pd catalysts [31–34] (Figure S7). In addition, the repeated experiment of CPT-1.5 was investigated in successive works. As described in Figure 4g, high yields of the target product remained after repeating the experiment five times, suggesting CPT-1.5 possessed excellent recyclability. The phase and morphology of CPT-1.5 after reuse were explored using XRD and TEM. As depicted in Figures S8 and S9, the main diffraction peaks of CPT-1.5 matched well with the original sample and its ordered mesoscopic structure remained good. In addition, the ICP tests showed that no obvious leaching of CuPd particles was found. These results showed that CPT1.5 had good structural stability, due to the strong interaction between CP particles and $TiO_2$, which could effectively prevent the leaching of CuPd during photocatalytic Stille reaction. The XPS test results showed two more peaks with low intensity that appeared at 335.5 and 341.5 eV for divalent Pd after CPT-1.5 was reused for five times (Figure S10), indicating that a fraction of the metallic Pd of the CuPd alloy on CPT-1.5 was oxidized to $Pd^{2+}$ in the repeated experiment. This might have been responsible for the tiny decline in the catalytic activity after repeating the test five times.

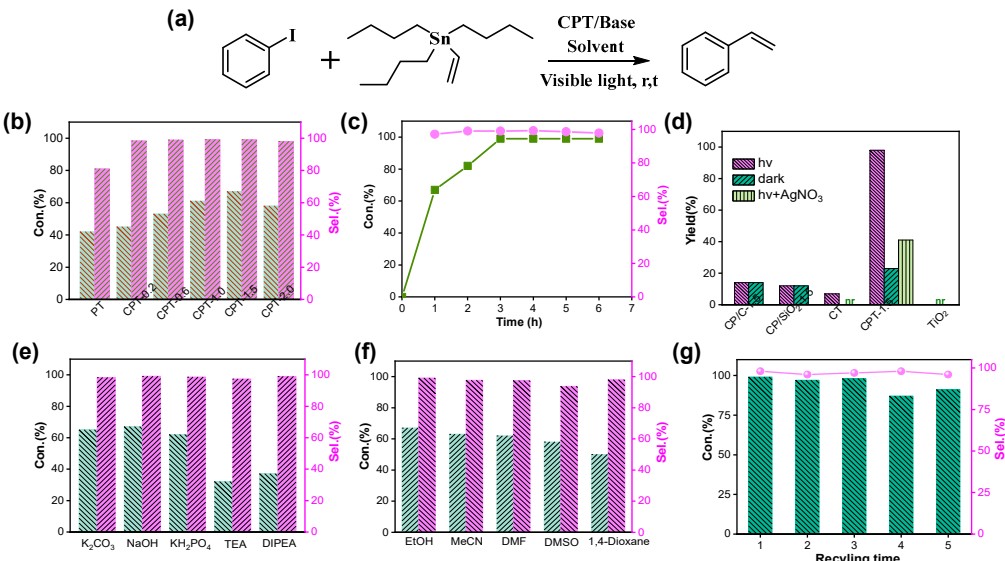

**Figure 4.** Photocatalytic activity of CPT under various conditions. (**a**) Reaction scheme for the photocatalyzed Stille coupling reaction. (**b**) Effect of the CuPd ratio on the photocatalytic activity of CPT. Effects of (**c**) time, (**d**) different catalysts, (**e**) bases, and (**f**) solvents on the photocatalytic activity. (**g**) Reusability of CPT-1.5 for the photocatalytic Stille coupling reaction.

To elucidate the mechanism responsible for the photocatalytic Stille coupling reaction, the influence of catalyst support was studied. As shown in Figure 4d, when $SiO_2$ or activated C was used as a catalyst support in place of $TiO_2$, the conversion of Iodobenzene was negligible. When silver nitrate was added as an electron trapping agent into the reaction of Iodobenzene with Tributyl(vinyl)tin to trap the electrons transferred from CuPd alloy to Iodobenzene, the yield of the photocatalytic Stille reaction decreased significantly, indicating that photoexcited electrons played a key role in this reaction. Liebeskind L.S. et al. proposed that the introduction of copper or copper salts could increase the concentration of tin reagent on the catalyst surface, thereby improving the conversion and selectivity of the Stille reaction [35]. Contrary to this, Simon P.H. et al. found that the introduction of copper or copper salt did not affect the oxidative addition of Pd during the Stille reaction. It only promoted the electron transfer of tin reagent and Iodobenzene, thereby improving the conversion of Iodobenzene [36]. Figure S11 shows the FR-IR spectrum of tin reagent adsorbing over CPT-1.5 and PT surfaces during the reaction process. There were two sets of absorption peaks that were located between 1250 and 1750 $cm^{-1}$ and could be assigned to the stretching vibration absorption peaks of C = C ($\nu$ = 1658 and 1619 $cm^{-1}$) and the bending vibration of C-H for $-CH_2$ ($\nu$ = 1403 $cm^{-1}$) and $-CH_3$ ($\nu$ = 1376 $cm^{-1}$), respectively. Additionally, the absorption peaks situated at 1006 and 997 $cm^{-1}$ could be originated from the bending vibration of C=C, and the peak at 828 $cm^{-1}$ could be assigned to the trans vibration of C-H for $-CH_2$. The lack of differentiation in the intensity of these characteristic peaks indicated that the concentration of tin reagent adsorbed on the surface of CPT-1.5 catalyst did not increase significantly. Therefore, the concentration variation in the tin reagent was not the main reason for the improved efficiency of the Stille reaction. Furthermore, the UV-Vis diffuse reflectance spectra of PT, CPT-1.5, and CPT-2.0 showed light absorption in the visible region, and their corresponding values of bandgaps were calculated to be 2.95, 2.78 and 2.8 eV, respectively, based on the Kubelka–Munk formula (Figures S12 and S13). Both CPT-1.5 and CPT-2.0 showed significant improvement in light-absorption in the visible region, which might be attributed to the synergistic effect of optimizing the energy band structure of CPT and the Cu plasma effect [37]. In addition, a control experiment was conducted by comparing the activity of $TiO_2$ with Cu and Pd doped alone (Figure 4b,d), which showed that the active center during the photocatalytic Stille reaction was mainly in the Pd sites. Furthermore, photocurrent and EIS test showed

that CPT possessed higher photocurrent density and smaller arc radius than those of PT under the same condition [38] (Figures S14 and S15). Additionally, PL spectral intensity of CPT was lower than that of PT (Figure S16). These results indicated that the modulation of CuPd ratio was critical to the efficient transfer of photoexcited charges, and it inhibited the recombination of electrons and holes [39].

Therefore, the optimal CuPd ratio could improve the light absorption of CPT on the one hand. On the other hand, with an increase in Cu ratio, the electron transfer from $TiO_2$ to Pd sites was accelerated, and the electron cloud density on the Pd surface increased [16]. These light-excited electrons were available at the surface of the Pd sites. Therefore, the Pd sites had a good affinity for both aryl halides and organostannanes, and the electrons at the Pd sites could also enhance their intrinsic ability to activate the reactant molecules.

To better understand the reaction mechanism of the photocatalytic Stille coupling reaction, the effects of bases were investigated (Figure 4e). Among the bases used, NaOH and $K_2CO_3$ were determined to be the most effective, whereas $KHPO_4$ was found to be less effective due to its weak alkalinity. This phenomenon was very similar to that of traditional Stille reaction. When organic bases such as TEA and DIPEA were used, an inferior conversion of Iodobenzene was found. This might be attributed to the strong electrostatic adsorption capacity of TEA and DIPEA, which were adsorbed on the surface of the Pd sites and inhibited the exposure of active sites, leading to the decline of catalytic activity [40].

Subsequently, the influences of different solvents on the Stille reaction catalyzed by CPT-1.5 under visible light were investigated. As shown in Figure 4f, non-polar organic solvent (1,4-Dioxane) exhibited a moderated activity under this type of photocatalytic Stille reaction. Generally, polar aprotic solvents were considered the ideal solvents for Stille reaction that could enable high yields of the desired products. However, except for MeCN, both DMF and DMSO gave unsatisfactory results when the reaction was carried out at room temperature under visible light, which was different from that of conventional Stille reaction. Notably, when EtOH was used as a solvent, a very high yield of the desired products was achieved under the same condition, which suggested that EtOH was more favorable for the photocatalytic Stille reaction. We speculated that EtOH might act as a hole trapping agent because it could be oxidized more readily by photogenerated hole transfer due to the relatively low oxidation potential, thus promoting the transfer of photogenerated electrons.

Based on the above experimental results, a possible mechanism responsible for the photocatalytic Stille coupling reaction that was catalyzed by CPT was proposed (Figure 5). Under visible light irradiation, the CPT hybrids were excited and the photogenerated electrons in the CB of the $TiO_2$ support could be transferred more easily to the CB of the CuPd alloy particles. Simultaneously, the photogenerated holes that remained in the VB of $TiO_2$ were transferred to the solvent. Therefore, the CPT hybrids efficiently facilitated the separation of photoinduced electron–hole pairs. In addition, the alloy nanoparticles could absorb visible light, and the generated hot electrons due to the plasma effect of Cu could become available at the surface of the Pd sites, allowing the Pd sites to have a good affinity for the reactant. These electron-rich Pd sites then underwent oxidative addition of the aryl halides, which was followed by transarylation of the organostannanes. Finally, the desired product was achieved after reductive elimination.

Finally, CPT was used to activate the photocatalytic Stille reaction of various aryl halides with organostannanes at room temperature (Table 1). Regardless of whether Tributyl(vinyl)tin or ributyl(phenyl)stannane was used, all substituents of the electron-donating group were effectively converted to their corresponding products in high yield. In addition, the steric hindrance of aryl halides rarely affected the yields of the Stille reaction. However, a slight decrease in product yields was found when the aryl halides belonged to the electron-withdrawing groups. This result verified that CPT hybrids could be applied to the photocatalytic Stille reaction with various aryl halides and organostannanes at room temperature.

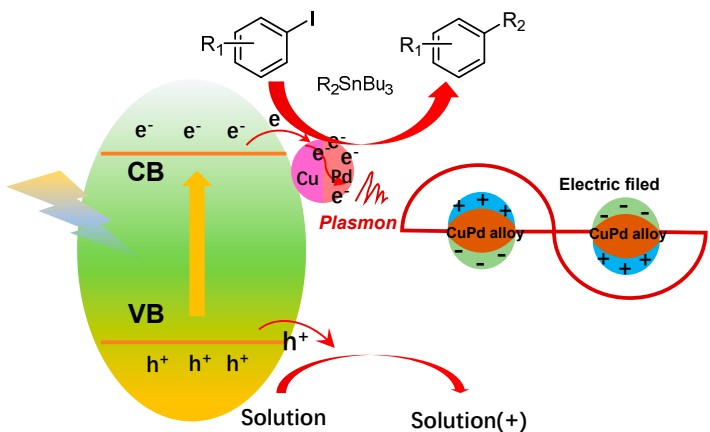

**Figure 5.** Proposed mechanism responsible for the photocatalytic Stille coupling reaction over CPT-1.5.

**Table 1.** Photocatalytic Stille Coupling Reactions of Various Aryl Halides over CPT-1.5.

| Enrty | Aryl halides | Product | Yield (%) |
|-------|-------------|---------|-----------|
| 1 | | | 98.2 |
| 2 | | | 97.3 |
| 3 | | | 96.6 |
| 4 | | | 97.1 |
| 5 | | | 95.4 |
| 6 | | | 86.4 |
| 7 | | | 90.1 |
| 8 | | | 88.2 |
| 9 | | | 85.7 |

Reaction conditions: Aryl Halides (0.1 mmol), CPT-1.5 (20 mg), $K_2CO_3$ (5 equiv), EtOH (2.0 mL), Xe lamp (300 W, $\lambda > 420$ nm), Time (3 h), rt, Determined by LC.

## 4. Conclusions

In summary, alloyed CuPd that is embedded in an ordered mesoporous $TiO_2$ has been rationally designed and synthesized using a pore-confined strategy for photocatalytic Stille coupling reaction under mild conditions. The atomic ratio and density of the CuPd on the mesoporous $TiO_2$ hybrids are readily controlled during synthesis, and we found that the CuPd ratio is the crucial factor for the photocatalytic activity of the CPT during Stille reaction under visible light. Moreover, solvent and base also have important influences on the efficiency of the photocatalytic Stille reaction. Experimental results show that the yields of target products are extraordinarily enhanced when protic solvents are used in the presence of $K_2CO_3$. In addition, CPT also shows good reusability and organic substrate compatibility with various aryl halides and organostannanes.

**Supplementary Materials:** The following supporting information can be downloaded at: https://www.mdpi.com/article/10.3390/catal12101238/s1, Figure S1: Ti 2p XPS spectra of CPT-1.5; Figure S2: O 2p XPS spectra of CPT-1.5; Figure S3: FESEM images of (a) CPT-0.2, (b) CPT-1.5 and (c) CPT-2.0; Figure S4: High-resolution FESEM images of (a) CPT-0.2, (b) CPT-1.5 and (c) CPT-2.0; Figure S5: High-resolution TEM images of (a,d) CPT-0.2, (b,e) CPT-1.0 and (c,f) CPT-2.0; Figure S6: The yield of styrene in the photocatalytic Stille reaction of Iodobenzene and Tributyl(vinyl)tin over different samples; Figure S7: Comparison of the activity of CPT-1.5 with those of previously reported photocatalysts for photocatalyzed stille reactons; Figure S8: XRD of CPT-1.5 before and after repeated test in photocatalytic Stille reaction; Figure S9: TEM of CPT-1.5 after repeated test in photocatalytic Stille reaction; Figure S10:Pd 3d XPS spectra of CPT-1.5 after repeated test in photocatalytic Stille reaction; Figure S11: FI-IR of tributylvinyltin over in different catalysts during the photocatalytic Stille reaction; Figure S12: UV-Vis DRS of PT, CPT-1.5 and CPT-2.0; Figure S13: The plots of transformed Kubelka-Munk function versus the energy of light; Figure S14: Transient photocurrent response of PT, CPT-1.5 and CPT-2.0; Figure S15: EIS Nyquist plots of PT, CPT-1.5 and CPT-2.0; Figure S16: PL curves of PT, CPT-1.5 and CPT-2.0; Table S1: Structural parameters of different samples; Table S2: Content of CuPd for different samples.

**Author Contributions:** Conceptualization, T.T. and L.J.; methodology, W.C. and J.S.; writing—original draft preparation, T.T. and L.J.; writing—review and editing, W.C. and J.S.; supervision, H.M. and Z.X.; funding acquisition, H.M. and Z.X. All authors have read and agreed to the published version of the manuscript.

**Funding:** This work was supported by the National Key Research and Development Program of China (2020YFA0211004), the National Natural Science Foundation of China (22176128, 21876114), Shanghai Government (21XD1422800, 19DZ1205102 and 19160712900), the Chinese Education Ministry Key Laboratory and International Joint Laboratory on Resource Chemistry, the Shanghai Eastern Scholar Program, the Shanghai Engineering Research Center of Green Energy Chemical Engineering (18DZ2254200), and the Shanghai Frontiers Science Center of Biomimetic Catalysis.

**Acknowledgments:** Thanks for the technical support from Hangzhou Normal University and Shanghai Institute of Technology.

**Conflicts of Interest:** The authors declare no conflict of interest.

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
