# Peer review of "Improving Photocatalytic Stille Coupling Reaction by CuPd Alloy-Doped Ordered Mesoporous TiO2"

_catalysts, doi:10.3390/catal12101238_

Round 1
Reviewer 1 Report
I have read carefully the submitted manuscript entitled “Improving Photocatalytic Stille Coupling Reaction by CuPd al- loy Doped Ordered Mesoporous TiO2. It is quite a good work and there are some minor issues required that need to be addressed before publication and hopefully, that will enhance the quality of the manuscript.
1- In section 2.4, the analytical method must be well detailed
2- In section 3.1, figures and tables that are in supplementary should be quoted and interpreted in detail, e.g. fig S6 , S7, S10, table S1 and S2.
3- For the reaction, is it that the reactive species generated by the catalyst in the presence of light does not degrade the reagents of the reaction? And how did you check that?
4- It is necessary to explain the mechanism of the reaction: transfer of electrons ....
5- it is necessary to calculate the energy gap to show that it is photoactive in the visible.
6- I propose to test the reaction in the presence of sunlight to evaluate the efficiency of the catalyst.
7- A comparison of the efficiency of the catalyst with the literature should be added
Author Response
Dear Reviewer,
Thanks for your comments on our manuscript. These comments do help us to improve the quality of our manuscript. All these comments have been fully considered during the revision of our manuscript. The related changes/additions have been added and highlighted with yellow color in the revised manuscript, and some references have been added,and we sincerely wish that our revisions and answers could be satisfied. We hope that these fully addressed revisions and explanations would make our work more comprehensible and readable. Thanks again for all your concern and kind support.
Comment 1:
In section 2.4, the analytical method must be well detailed
Response:
According to the comments of reviewer, the analytical method in section 2.4 was described in detailed and highlighted with yellow color in the revised manuscript.
“After reaction for a certain period time, the reaction mixture was filtered and analyzed by high performance liquid chromatograph (Agilent 6410 Series Triple Quad), equipped with an Agilent C18 reverse-phase chromatographic column and UV detector (wavelength 280 nm), and the mobile phase was 55:45 acetonitrile/water (flow rate 1 mL/min, injection volume 5 μL). The conversion of aryl halides and the yield of corresponding products were determined by using trimethylbenzene as the internal standard sample.” (page 3, line 107-113)
Comment 2:
In section 3.1, figures and tables that are in supplementary should be quoted and interpreted in detail, e.g. fig S6, S7, S10, table S1 and S2.
Response:
Thanks for the helpful suggestion. The figures and tables in supporting information have been well quoted and interpreted, and their descriptions have been added in the revised manuscript accordingly.
“In addition, pore volumes and BET surface areas of CPT-0.2, CPT-1.5 and CPT-2.0 were estimated to be 0.46, 0.32 and 0.30 cm3/g, and 180, 164 and 155 m2/g with a pore size distribution of 4.1-3.2 nm, respectively (Tab.S1). The high BET specific surface areas, large pore volumes and uniform pore size distribution (Fig.2d, Tab.S1) for all of CPT, determining the variation in atom ratio of CuPd slightly effect their mesostructured.”(page 3, line 126-131)
“There are two sets absorption peaks located between 1250 and1750 cm-1 can be assigned to the stretching vibration absorption peaks of C=C (ν=1658 and 1619 cm-1) and the bending vibration of C-H for -CH2 (ν=1403 cm-1) and -CH3 (ν=1376 cm-1) respectively. Additionally, the absorption peaks situated at 1006 and 997 cm-1 should be originated from the bending vibration of C=C, and the peak at 828 cm-1 should be assigned to trans vibration of C-H for -CH2. No distinguish in the intensity of these characteristic peaks indicated the concentration of tin reagent adsorbed on the surface of CPT-1.5 catalyst insignificantly increase.”(page 5-6, line 210-217)
“the UV-Vis diffuse reflectance spectra of PT, CPT-1.5 and CPT-2.0 showed light-absorption in the visible region, and their corresponding values of bandgaps were calculated to be 2.95, 2.78 and 2.8eV, respectively, based on the Kubelka–Munk formula. Both CPT-1.5 and CPT-2.0 showed significantly improved light-absorption in the visible region, which may be attributed to the synergistic effect of optimizing the energy band structure of CPT and Cu plasma effect” (page 6, line 219-224)
“Furthermore, photocurrent and EIS test shows CPT possess higher photocurrent density and smaller arc radius than that of PT under the same conditions (Fig.S14,15). Also, PL spectral intensity of CPT were lower than that of PT(Fig.S16). These results indicated that the modulation of CuPd ratio was critical to the efficient transfer of photoexcited charge, and inhibit the recombination of electrons and holes” (page 6, line 227-231)
Comment 3:
For the reaction, is it that the reactive species generated by the catalyst in the presence of light does not degrade the reagents of the reaction? And how did you check that?
Response:
Thank you very much for the valuable opinion, which is very helpful for us to understand the reaction process deeply. In fact, the production of active oxygen species is often involved in photocatalytic organic reactions, and the organic reaction substrates should inevitably be degraded, our reaction system is no exception. However, due to the complexity of asymmetric C-C coupling reaction, it is possible to generate a variety of reaction intermediates, so we cannot determine the amount of degradation of the reagents in this reaction system. Despite all this, we also have reason to think that the reactants degraded in the reaction process can be ignored and without affecting the conversion and selectivity of this reaction. This mainly depends on two aspects: on the one hand, the concentration of Iodobenzene is 0.05 mmol/ml (10000 ppm/L) which is nearly 1000 times higher than the concentration applicable to organic degradation. On the other hand, ethanol used as solvent in this reaction can also serve as hole trapping agents to inhibit the degradation of reaction substrate.
Comment 4:
It is necessary to explain the mechanism of the reaction: transfer of electrons ...
Response:
The possible mechanisms mechanism responsible for transfer of electrons and photocatalytic Stille coupling reactions have been proposed and the relative description was presented in revised manuscript.
Based on the above experimental results, a possible mechanism responsible for photocatalyzed Stille coupling reactions catalyzed by CPT was proposed (Fig.5). Under visible light irradiation, the CPT hybrids are excited, the photogenerated electrons in the CB of the TiO2 support could be transferred easily to the CB of the CuPd alloy particles. Simultaneously, photogenerated holes remained in the VB of TiO2 are transferred to the solvent. Therefore, the CPT hybrids efficiently facilitate the separation of photoinduced electron–hole pairs. In addition, the alloy nanoparticles could absorb visible light, and generate hot electrons due to the plasma effect of Cu, which could be available at the surface Pd sites and allow Pd sites have good affinity for the reactant. These electron-rich Pd sites then undergo oxidative addition of an aryl halide, followed by transarylation of the organostannanes. Finally, the desired product was achieved after reductive elimination. (Page 8, line 273-282)
Figure 5. Proposed mechanism responsible for photocatalytic Stille coupling reactions over CPT-1.5.
Comment 5:
It is necessary to calculate the energy gap to show that it is photoactive in the visible.
Response:
The corresponding values of bandgaps for PT, CPT-1.5 and CPT-2.0 were calculated to be 2.95, 2.78 and 2.8eV, respectively, based on the Kubelka–Munk formula. It indicates that CPT can be excited by visible light and catalyze the subsequent stille reaction. The related illustrations have been added in the revised manuscript and the plots of transformed Kubelka-Munk function was presented in supporting information as Fig. S13.
“Furtherly, the UV-Vis diffuse reflectance spectra of PT, CPT-1.5 and CPT-2.0 showed light-absorption in the visible region, and their corresponding values of bandgaps were calculated to be 2.95, 2.78 and 2.8eV, respectively, based on the Kubelka–Munk formula(Fig.S12,13). Both CPT-1.5 and CPT-2.0 showed significantly improved light-absorption in the visible region, which may be attributed to the synergistic effect of optimizing the energy band structure of CPT and Cu plasma effect” (Page 6, line 219-224)
Figure S13. The plots of transformed Kubelka-Munk function versus the energy of light.
Comment 6:
I propose to test the reaction in the presence of sunlight to evaluate the efficiency of the catalyst.
Response:
Thanks for the helpful and valuable suggestion. We tried to use sunlight to catalyze this reaction. Unfortunately, CPT does not show any activity under the radiation of sunlight, the absorption and effective utilization of sunlight are still limited. We hope to develop more efficient and cheaper photocatalysts for solar induced C-C coupling reaction in future research.
Comment 7:
A comparison of the efficiency of the catalyst with the literature should be added.
Response:
Thanks very much. According to the suggestion, the efficiency of CPT-1.5 have been compared with previously reported heterogeneous Pd-based catalysts for C-C cross-couplings in the identical conditions. The corresponding result was presented in supporting information as Fig. S7.
Figure S7. Comparison of the activity of CPT-1.5 with those of previously reported photocatalysts for photocatalyzed stille reactons.

Reviewer 2 Report
In this work, alloyed CuPd (CP) nanoclusters were anchored on ordered mesoporous TiO2(CPT), and the resultant CPT exhibited extraordinary photo-catalytic activity in Stille reactions under visible light. I think it is acceptable after the following concerns are addressed.
1) The samples were labelled as CPT-1.0, CPT-1.5, CPT-2.0, etc. The meaning of the numbers after CPT needs to be explained.
2) In Fig. S9, the equivalent circuit diagram of the test needs to be added (Chem. Eng. J. 2022, 431, 133446). Furthermore, the data obtained should be displayed in dots, not connected by dots and lines.
3) In Figure S7, it is suggested to mark the values of ordinate.
4) From Fig. 4g, it can be seen that the yield of the target product declined at the 4 times repeated experiment, suggesting the stability of CPT-1.5 is not excellent. A reasonable explanation is needed.
5) The “pore confinement effect” was mentioned in abstract, which is suggested to be introduced in Introduction (ACS Nano 2021, 15, 6551-6561).
6) The catalytic performance of the developed CPT is expected to compare with those similar catalysts reported in literatures.
7) Some related papers can be referred to, such as Catalysts 2016, 6, 117-127; etc.
Author Response
Dear Reviewer,
Thanks for your comments on our manuscript. These comments do help us to improve the quality of our manuscript. All these comments have been fully considered during the revision of our manuscript. The related changes/additions have been added and highlighted with yellow color in the revised manuscript, and some references have been added,and we sincerely wish that our revisions and answers could be satisfied. We hope that these fully addressed revisions and explanations would make our work more comprehensible and readable. Thanks again for all your concern and kind support.
Comment 1:
The samples were labelled as CPT-1.0, CPT-1.5, CPT-2.0, etc. The meaning of the numbers after CPT needs to be explained.
Response:
Thanks for valuable suggestion, the meaning of the numbers after CPT have been explained in the revised manuscript.
“Similar procedures were employed to fabricated Cu-TiO2 and CuPd-TiO2, except that Pd(OAc)2 was replaced in whole or in part by Cu(OAc)2, the obtained sample were denoted as CT and CPT-X, respectively. Where, X was the molar ratio of Cu to Pd. In this paper, CPT-0.2, CPT-0.6, CPT-1.0, CPT-1.5 and CPT-2.0 were synthesized and their photocatalytic properties were studied, and the molar ratio of Cu to Pd for each sample was determined by ICP and XPS (Tab.S2). ”(page 6, line 97-102)
Comment 2:
In Fig. S9, the equivalent circuit diagram of the test needs to be added (Chem. Eng. J. 2022, 431, 133446). Furthermore, the data obtained should be displayed in dots, not connected by dots and lines.
Response:
Thanks for the helpful suggestion, the impedance diagram has been redrawn as suggested and added into supporting information as Fig. S15
Figure S15. EIS Nyquist plots of PT, CPT-1.5 and CPT-2.0
Comment 3:
In Figure S7, it is suggested to mark the values of ordinate.
Response:
The values of ordinate in Figure S7 (Figure S12 in the revised supporting information) have been marked.
Figure S12. UV-Vis DRS of PT, CPT-1.5 and CPT-2.0
Comment 4:
From Fig. 4g, it can be seen that the yield of the target product declined at the 4 times repeated experiment, suggesting the stability of CPT-1.5 is not excellent. A reasonable explanation is needed.
Response:
Thanks for this constructive suggestion, we have studied the structure, alloy content and chemical structure of CPT-1.5 after reused for 5 times by XRD,TEM and XPS, the phase and morphology of CPT-1.5 was well kept and no obvious alloy leaching was found, this shows that the stability of CPT is very good. However, the XPS test results showed that a small amount of bivalent palladium appeared after repeated test. We speculate this may be the reason why the catalytic activity decreased slightly after four times of application. The XRD, TEM and XPS spectrum was displayed in supporting information as Fig. S8-10, and the relative analysis of this results was documented in the revised manuscript.
“The phase and morphology of CPT-1.5 after reused were explored by XRD, TEM. As depicted in Fig. S8 and Fig. S9. The main diffraction peaks of CPT-1.5 match well with the original sample and its ordered mesoscopic structure remains good. In addition, the ICP tests showed that no obvious leaching of CuPd particles is found. These results show that CPT1.5 has good structural stability, due to the strong interaction between CP particles and TiO2, which can effectively prevent the leaching of CuPd in the photocatalyzed stille reaction. XPS test results show two more peaks with very low intensity appeared at 335.5 and 341.5 eV for divalent Pd after CPT-1.5 reused for 5 times (Fig.S10), indicated that fraction of metallic Pd of CuPd alloy was oxidized to Pd2+ in the repeated experiment, which may be responsible for the tiny decline in the catalytic activity after 5 times repeated test.” (page 5, line 186-196)
Figure S8. XRD of CPT-1.5 before and after repeated test in photocatalytic Stille reaction.
Figure S9. TEM of CPT-1.5 after repeated test in photocatalytic Stille reaction
Figure S10. Pd 3d XPS spectra of CPT-1.5 after repeated test in photocatalytic Stille reaction
Comment 5:
The “pore confinement effect” was mentioned in abstract, which is suggested to be introduced in Introduction (ACS Nano 2021, 15, 6551-6561).
Response:
As reference suggested, the “pore confinement effect” was introduced in introduction.
“In this process, the uniform P123/ titanium oligomer composite micelles were formed and worked as the building blocks to enable the self-assembly of micelles into ordered mesostructured, where the oligomer not only provides reinforced structure to retain highly ordered mesostructure, but also acts as cross-linked networks to keep confinement of other metal ions. After crystallization, the isolated CuPd atoms are confined in the pore wall of mesostructured TiO2” (page 2, line 64-69)
Comment 6:
The catalytic performance of the developed CPT is expected to compare with those similar catalysts reported in literatures.
Response:
The efficiency of CPT-1.5 have been compared with previously reported heterogeneous Pd-based catalysts for C-C cross-couplings in the identical conditions. The corresponding result was presented in supporting information as Fig. S7.
Figure S7. Comparison of the activity of CPT-1.5 with those of previously reported photocatalysts for photocatalyzed stille reactons.
Comment 7:
Some related papers can be referred to, such as Catalysts 2016, 6, 117-127; etc.
Response:
The relative references have been cited as 21, 24, 31, 32, 33, 34, and 38 in the revised manuscript.

Reviewer 3 Report
The manuscript deals with preparation of heterogeneous catalysts consisting of Cu-Pd nanoparticles on ordered mesoporous TiO2 and their photocatalytic activity in Stille coupling reaction. All experiments and samples’ characterization are well designed and performed, the results are very well described and interpreted.
Below are my comments that do not shadow my positive evaluation.
Line 18: perhaps “atomic ratio of CP” not CT
Line 93-94: What are the values for X and how many samples were produced? Please, here introduce the series studied and samples labels. So that, the reader to know what samples are under investigation.
Line 95 and elsewhere: Stiller – please, delete “r”
Line 98: Is it Xenon lamp? Please correct accordingly.
Line 102: As can be seen at the end of the manuscript, not only iodobenzene was reacted and thus other biaryl products were produced as well. Please, edit the sentence with correct info.
Line 126: Figure 2 Caption: (e) and (f) are in opposite order - (e) for Pd and (f) for Cu
Line 132-133: the XPS peaks are mixed up - 2p1/2 is at 952.1 eV.
Line 166: “the photocatalytic activity of CPT increased first and then slightly decreased with the amount of Cu increas ing, which presented a volcano-type trend with the increase of CuPd ratio.” In fact, it is not visible in the Figure 4b. Please, provide more data or another figure.
Line 188-189: I am afraid that there is no band for –CH3 at 1750 cm-1. Please, give a reference for stretching vibration of -CH3 at 1750 cm-1 and for C=C at 1250 cm-1.
Line 195: Do you mean 4b and 4d?
Line 225: Acetonitrile is a polar aprotic solvent and dioxane is non-polar. Please, check again properties of used solvents and reconsider the conclusions made in this paragraph.
Line 240: I would recommend this table to be shown in the main body not in SI, as the experiments demonstrate the catalytic activity of CPT with other reagents as well, which enhance the scopes of the work.
Author Response
Dear Reviewer,
Thanks for your comments on our manuscript. These comments do help us to improve the quality of our manuscript. All these comments have been fully considered during the revision of our manuscript. The related changes/additions have been added and highlighted with yellow color in the revised manuscript, and some references have been added,and we sincerely wish that our revisions and answers could be satisfied. We hope that these fully addressed revisions and explanations would make our work more comprehensible and readable. Thanks again for all your concern and kind support.
Comment 1:
Line 18: perhaps “atomic ratio of CP” not CT
Response:
Thanks for you careful reading and friendly reminders. Your suggestions are valuable for improving the preciseness and logic of our article. We have carefully checked this manuscript and corrected the spelling and expression errors one by one.
Comment 2:
Line 93-94: What are the values for X and how many samples were produced? Please, here introduce the series studied and samples labels. So that, the reader to know what samples are under investigation.
Response:
According to this suggestion, we have detailed the preparation information of the catalyst in section 2.3.
“Similar procedures were employed to fabricated Cu-TiO2 and CuPd-TiO2, except that Pd(OAc)2 was replaced in whole or in part by Cu(OAc)2, the obtained sample were denoted as CT and CPT-X, respectively. Where, X was the molar ratio of Cu to Pd. In this paper, CPT-0.2, CPT-0.6, CPT-1.0, CPT-1.5 and CPT-2.0 were synthesized and their photocatalytic properties were studied, and the molar ratio of Cu to Pd for each sample was determined by ICP and XPS (Tab.S2).” (page 6, line 97-102)
Comment 3:
Line 95 and elsewhere: Stiller – please, delete “r”
Response:
All the Stiller involved in this manuscript have been corrected to Stille.
Comment 4:
Line 98: Is it Xenon lamp? Please correct accordingly.
Response:
Have been corrected.
Comment 5:
Line 102: As can be seen at the end of the manuscript, not only iodobenzene was reacted and thus other biaryl products were produced as well. Please, edit the sentence with correct info.
Response:
Thanks for this valuable suggestion, this sentence has been re-edited.
“CPT (20 mg) were dispersed in 2 mL EtOH, followed by addition of aryl halides (0.1 mmol), organostannanes (C14H30Sn or C18H32Sn) and 0.5 mmol of K2CO3 in a Pyrex reactor equipped with a rubber septum. The reaction solution was irradiated with a 300 W Xenon lamp (λ > 420 nm) at room temperature.”(page 3, line 103-106)
Comment 6:
Line 126: Figure 2 Caption: (e) and (f) are in opposite order - (e) for Pd and (f) for Cu
Line 132-133: the XPS peaks are mixed up - 2p1/2 is at 952.1 eV.
Response:
Thanks for this valuable suggestion, we have carefully checked and corrected the expression errors.
Comment 7:
Line 166: “the photocatalytic activity of CPT increased first and then slightly decreased with the amount of Cu increasing, which presented a volcano-type trend with the increase of CuPd ratio.” In fact, it is not visible in the Figure 4b. Please, provide more data or another figure.
Response:
Thanks very much, the figure for yield of styrene was offered in supporting information (FigureS6), where the trend of volcanic type for the photocatalytic activity of CPT is obvious with the increase of CuPd ratio.
Figure S6. The yield of styrene in the photocatalytic Stille reaction of Iodobenzene and Tributyl(vinyl)tin over different samples
Comment 8:
Line 188-189: I am afraid that there is no band for –CH3 at 1750 cm-1. Please, give a reference for stretching vibration of -CH3 at 1750 cm-1 and for C=C at 1250 cm-1.
Response:
Thanks for your valuable comments, we have carefully analyzed the infrared results and revised the inaccurate conclusions in the revised version.
“There are two sets absorption peaks located between 1250 and1750 cm-1 can be assigned to the stretching vibration absorption peaks of C=C (ν=1658 and 1619 cm-1) and the bending vibration of C-H for -CH2 (ν=1403 cm-1) and -CH3 (ν=1376 cm-1) respectively. Additionally, the absorption peaks situated at 1006 and 997 cm-1 should be originated from the bending vibration of C=C, and the peak at 828 cm-1 should be assigned to trans vibration of C-H for -CH2. No distinguish in the intensity of these characteristic peaks indicated the concentration of tin reagent adsorbed on the surface of CPT-1.5 catalyst insignificantly increase. ”(page5-6, line 210-217)
Comment 9:
Line 195: Do you mean 4b and 4d?
Response:
Yes, it has been corrected.
Comment 10:
Line 225: Acetonitrile is a polar aprotic solvent and dioxane is non-polar. Please, check again properties of used solvents and reconsider the conclusions made in this paragraph.
Response:
Thanks very much, we have checked the properties of the solvents used and reasonably analyzed the experimental results.
“Subsequently, the influences of different solvents on the Stille reaction catalyzed by CPT-1.5 under visible light were investigated, as showed in Fig. 4f, non-polar organic solvent (1,4-Dioxane) exhibited moderated activity in this photocatalytic Stille reaction. Generally, polar aprotic solvents were considered as the ideal solvents for Stille reactions to enable high yields of the desired products. However, except MeCN, both DMF and DMSO gave unsatisfactory results when the reaction was carried out at room temperature under visible light, which is different from the that of conventional Stille reactions. Notably, when EtOH was used as solvent, a very high yield of the desired product was achieved at the same condition, which suggested that EtOH is more favorable for the photocatalyzed Stille reactions. We speculated that EtOH may act as hole trapping agent, because it could be oxidized more readily by photogenerated hole transfer due to the relatively low oxidation potential, thus promoted the transfer of photogenerated electrons.”(page 7, line 251-262 )
Comment 11:
Line 240: I would recommend this table to be shown in the main body not in SI, as the experiments demonstrate the catalytic activity of CPT with other reagents as well, which enhance the scopes of the work.
Response:
Thanks very much. According to the suggestion, the table showed in SI have been re-located in the revised manuscript as Tab1.

Round 2
Reviewer 1 Report
The revised version of the manuscript may acceptable to the journal standard.